# Impact of Neutropenia on Clinical Outcomes after Lung Transplantation

**DOI:** 10.3390/medsci12040056

**Published:** 2024-10-16

**Authors:** Raquel Sanabrias Fernández de Sevilla, Ana Concepción Sánchez Cerviño, Rosalía Laporta Hernández, Myriam Aguilar Pérez, Christian García Fadul, Sarela García-Masedo Fernández, Amelia Sánchez Guerrero, María Piedad Ussetti Gil

**Affiliations:** 1Hospital Pharmacy Department, Hospital Universitario Puerta de Hierro, 28222 Majadahonda, Spain; raquel.sanabrias@salud.madrid.org (R.S.F.d.S.); anaconcepcion.sanchez@salud.madrid.org (A.C.S.C.); asguerrero@salud.madrid.org (A.S.G.); 2Pulmonology Department, Hospital Universitario Puerta de Hierro, 28222 Majadahonda, Spain; rosalia.laporta@salud.madrid.org (R.L.H.); myriam.aguilar@salud.madrid.org (M.A.P.); christian.garcia@salud.madrid.org (C.G.F.); 3Microbiological Department, Hospital Universitario Puerta de Hierro, 28222 Majadahonda, Spain; sarelagmf@gmail.com

**Keywords:** neutropenia, granulocyte colony-stimulating factors, lung transplantation, solid organ transplant, outcomes, allograft dysfunction, overall survival, infections

## Abstract

Background/Objectives: Neutropenia is a frequent complication among solid organ transplant (SOT) recipients receiving immunosuppressive therapy and antimicrobial prophylaxis. However, there are limited studies analysing the frequency and impact of neutropenia in lung transplant recipients (LTRs). Our aim was to analyse the frequency of neutropenia, the need for granulocyte colony-stimulating factor (GCSF) treatment within the first 18 months post-transplant and its association with acute rejection, chronic lung allograft dysfunction (CLAD), overall survival and the development of infections. Methods: This observational and retrospective study recruited 305 patients who underwent lung transplantation between 2009 and 2019, with outpatient quarterly follow-up during the first 18 months post-surgery.Results: During this period, 51.8% of patients experienced at least one episode of neutropenia. Neutropenia was classified as mild in 50.57% of cases, moderate in 36.88% and severe in 12.54%. GCSF treatment was indicated in 23.28% of patients, with a mean dose of 3.53 units. No statistically significant association was observed between neutropenia or its severity and the development of acute rejection, CLAD or overall survival. However, the patients who received GCSF treatment had a higher mortality rate compared to those who did not. Sixteen patients (5.25%) developed infections during neutropenia, with bacterial infections being the most common. Conclusions: Neutropenia is common in the first 18 months after lung transplantation and most episodes are mild. We did not find an association between neutropenia and acute rejection, CLAD, or mortality. However, the use of GCSF were associated with worse post-transplant survival.

## 1. Introduction

Neutropenia is a common complication in solid organ transplant (SOT) recipients due to immunosuppressive therapy and antimicrobial prophylaxis. This haematological toxicity has been associated with drugs such as mycophenolate mofetil, valganciclovir and sulfamethoxazole-trimethoprim [1,2,3]. Although there are currently no specific management protocols for neutropenia in SOT [4], in clinical practice, these drugs are often temporarily discontinued until the neutrophil counts recover, which may result in negative consequences for the patient, such as severe infections, graft rejection or even death [4,5,6].

A systematic review of 82 studies on the impact of leukopenia and neutropenia in kidney transplant recipients found an association between neutropenia and the development of acute rejection and opportunistic infections. However, the results were inconclusive in terms of graft loss and mortality. The authors emphasised the importance of early interventions to reduce the risk of haematological toxicity after transplantation [7].

There are few studies analysing the frequency and impact of neutropenia in lung transplant recipients (LTRs). A retrospective study involving 228 LTRs showed a higher mortality in the group of patients with severe neutropenia [8].

Granulocyte colony-stimulating factors (GCSFs) are used to reverse neutropenia and prevent infections. The administration of CGSFs in transplant recipients is very common and has been considered safe [6,9]. Moreover, its use in heart transplant recipients has been associated with a reduction in acute rejection events and graft vasculopathy, suggesting its potential immunomodulatory effect [9]. However, in lung and kidney transplant recipients, its administration has been associated with an increased risk of acute and chronic rejection [7,8,10].

Our aim was to analyse the frequency of neutropenia and the need for treatment with GCSF in the first 18 months post-transplant in LTRs and its association with overall survival, acute rejection (AR), chronic lung allograft dysfunction (CLAD) and the development of infections.

## 2. Methods

This observational and retrospective study includes patients who received their first lung transplant between 2009 and 2019, with a protocolled outpatient follow-up during the first 18 months post-surgery.

After surgery, patients were monitored in the outpatient clinic of the Department of Pulmonology. The follow-up protocol includes a medical visit, routine analysis with immunosuppressant levels and spirometry. The frequency of visits varies depending on the time post-transplant: one week after hospital discharge, fortnightly for the first month, monthly for the next three months, bimonthly for the first year and quarterly for life.

The immunosuppression protocol in our centre includes tacrolimus (plasma levels between 5–12 ng/mL depending on the time post-transplant), mycophenolate mofetil (500–1000 mg BID) and prednisone in decreasing doses up to 0.5 mg/kg every 48h for the first 12 months post-transplant.

Sulfamethoxazole-trimethoprim prophylaxis was initiated at hospital discharge and is maintained for life in the absence of haematological and/or digestive toxicity.

Patients at risk of cytomegalovirus (CMV) infection (R+ and D+/R−) received prophylaxis with intravenous ganciclovir followed by oral valganciclovir adjusted for renal function for the first 6–12 months post-transplant. High risk patients (D+/R−) also received CMV hyperimmune gamma globulin 100 mL monthly during the first year.

Immunosuppression management during neutropenia was based on the patient’s characteristics and the balance between the risk of rejection or infection. Mycophenolate was reduced or temporarily discontinued, while the usual doses of tacrolimus and corticosteroids were maintained. In relation to antimicrobial prophylaxis, sulfamethoxazole/trimethoprim and valganciclovir were also discontinued, and we used specific CMV hyperimmunoglobulin as an alternative for CMV prophylaxis.

The criteria for prescribing GCSF were based on the severity of the neutropenia, the patient’s characteristics and the physician’s criteria.

Patient data were obtained from the electronic medical records. The variables included in this study were age at transplantation, sex, baseline disease, date and type of transplantation, CMV serology (donor and recipient), neutropenia and its severity (mild: 1500–1000/mL, moderate: 1000–500/mL, severe: <500/mL), administration of GCSF (date and dose), concomitant infection (type and microorganism), acute rejection (date and severity), chronic lung allograft dysfunction (date), overall survival.

We performed a descriptive statistical analysis of qualitative variables. The association between neutropenia and the use of GCSF with AR, CLAD and survival was analysed using logistic regression and the Fine–Gray model.

## 3. Results

We included 305 patients in the study, 197 (64.59%) of whom were male, with a mean age at the time of transplantation of 53.3 years (range 16.3–68.2). The most frequent underlying disease was chronic obstructive pulmonary disease (42.30%), followed by interstitial lung disease (37.7%), cystic fibrosis (14.43%), bronchiectasis (2.95%) and others (2.62%). Most patients (83.61%) had positive serology for cytomegalovirus (R+) at the time of transplantation (Table 1).

We observed at least one episode of neutropenia during the first 18 months post-transplant in 154 patients (51.8%). Neutropenia was recurrent in 35.38% of recipients. Neutropenia was detected in 263 of the 1554 visits made (16.9%) during the study period. We found no significant differences in the occurrence of neutropenia when considering the time post-transplantation across different periods. The severity of neutropenia was classified as mild in 50.57% of cases (N = 133), moderate in 36.88% (N = 97) and severe in 12.54% (N = 33).

No statistical significant association was observed between the development of neutropenia and AR (*p* = 0.7053, CLAD (*p* = 0.1896) or overall survival (*p* = 0.7557) (Figure 1). We also found no significant association between the severity of neutropenia, AR (*p* = 0.7053), CLAD (*p* = 0.1896) or survival (*p* = 0.2418) (Figure 2).

Seventy-one patients (23.28%) were treated with GCSF, and the mean dose administered per patient was 3.53 units (SD ± 4.2). The severity of neutropenia was mild in 18 patients and moderate/severe in 53. Regardless of its severity, GCSF administration was prescribed more frequently in patients prone to bacterial or fungal infections after lung transplantation.

Sixteen patients (5.25%) developed infections during neutropenia, the most frequent being bacterial (Table 2). The locations of infection were pulmonary (N = 9; 56%), gastrointestinal (N = 2; 12.5%), sepsis (N = 2; 12.5%) and others (N = 2; 12.5%).

We did not observe a statistically significant relationship between GCSF administration and the development of AR (*p* = 0.2121) or CLAD (*p* = 0.8642). However, patients who received GCSF showed higher mortality than patients who did not (HR 2.30; 95% CI 1.32–4.00; *p* = 0.0025) (Figure 3) [11]. The causes of death in these patients were infections in 10 (55.60%), progressive CLAD in 5 (27.78%), humoral rejection in 2 (11.11%) and massive haemoptysis in 1 (5.56%).

In our study, we observed that half of the LTRs experienced at least one episode of neutropenia during the first 18 months post-transplantation, and that neutropenia may be recurrent in one third of them. The frequency of neutropenia in LTRs is higher than that described in other SOT recipients. In heart transplant recipients, a percentage of neutropenia ranging from 21 to 30% [12,13,14] has been reported, while in kidney transplant recipients, the frequency of neutropenia varies widely, from 15 to 58% [15]. These wide oscillations are probably related to differences in immunosuppression schemes and antimicrobial prophylaxis used in the different organs and transplantation units.

In relation to LTRs, the observed frequency of 51.3% in our study was higher than the 44.3% previously described by Tague et al. in their cohort of 228 recipients. This difference could be attributed to variations in follow-up time. However, most of the episodes we detected were mild or moderate, and only 12.5% were considered severe. This may be related to the frequent monitoring of patients in our unit and the early use of GCSF to prevent the development of infectious complications [5,16].

Severe neutropenia has been associated with an increased risk of infection and worse survival in a series of lung [8] or kidney transplant recipients [16]. However, in our series, we observed no association between the presence of neutropenia, acute rejection, CLAD and patient survival. The low frequency of severe neutropenia observed by us and the frequent and early use of GCSF may partly justify this lack of statistical association. The administration of GCSF has been associated with the development of acute graft rejection [10]. However, in our series, we did not observe an association between the use of GCSF and the development of AR or CLAD. Nevertheless, our patients who received GCSF had a higher mortality rate. This finding suggests that patients requiring GCSF therapy have a worse long-term prognosis.

Bacterial infections are common in neutropenic patients [17]. LTRs frequently develop pulmonary bacterial infections as a consequence of the characteristics of the graft and the intense immunosuppression treatment they receive. However, in our study, we observed a low number of infections during episodes of neutropenia. This low frequency may be related to the close follow-up these recipients undergo in our unit and to the early administration of GCSF.

The main limitations of our study are its single-centre retrospective nature and the follow-up period limited to the first 18 months post-surgery. Although this period is associated with the highest risk of post-transplant complications, the interval analysed may be insufficient for a comprehensive and extrapolatable assessment of long-term outcomes. Extending the follow-up period would provide a more comprehensive and accurate picture of the evolution of patients, especially with the progressive increase in post-transplant survival.

## 4. Conclusions

In conclusion, we have observed that neutropenia is very frequent in the first 18 months post-lung transplantation. Most episodes of neutropenia -were mild, but they may be persistent in more than one third of patients. We have not observed an association between the presence of neutropenia and acute rejection and CLAD or mortality. However, we have observed that the use of GCSF were associated with worse post-transplant survival.

## Figures and Tables

**Figure 1 medsci-12-00056-f001:**
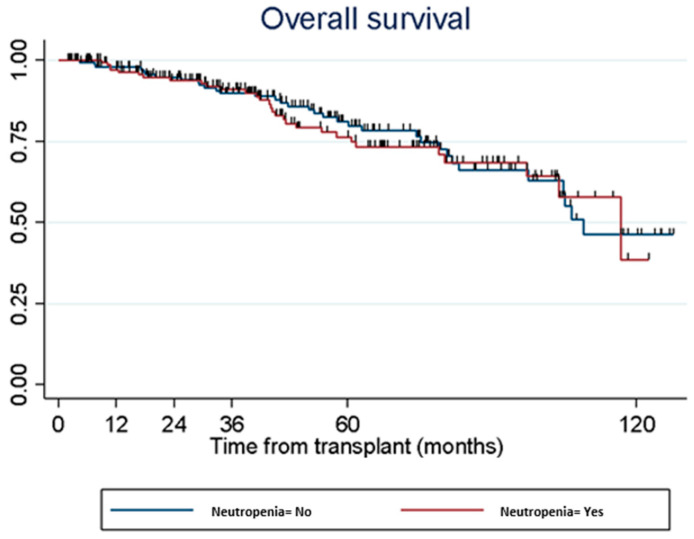
Association of neutropenia with overall survival.

**Figure 2 medsci-12-00056-f002:**
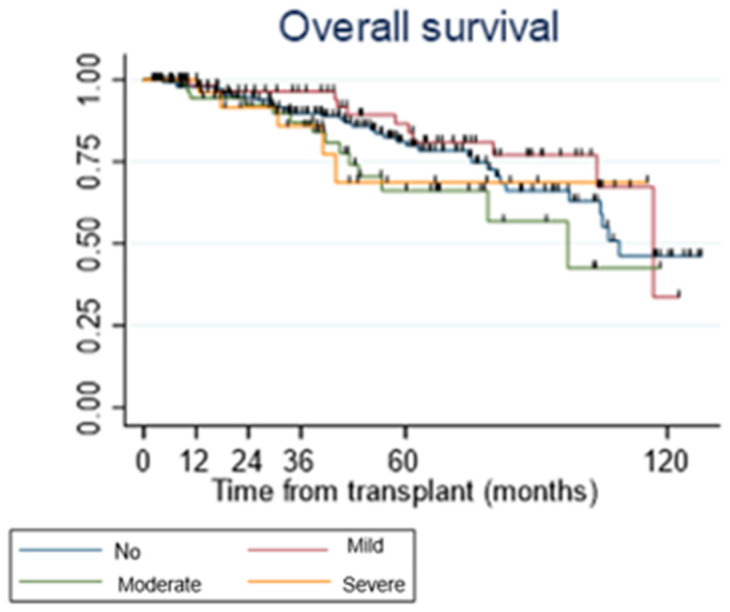
Overall survival according to neutropenia severity.

**Figure 3 medsci-12-00056-f003:**
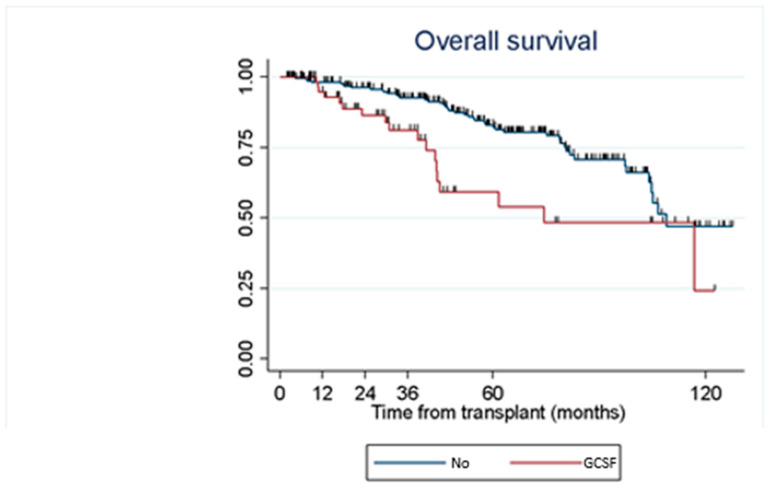
Association of need for treatment with GCSF and overall survival [11].

**Table 1 medsci-12-00056-t001:** Demographic and clinical variables.

	N = 305	N = 71 (GCSF-Treated)
**Recipient Age (years, mean (range))**	53.3 (16.2–68.2)	53.4 (20.4–66.0)
Recipient Gender (n, %)		
Female	108 (35.41)	23 (32.39)
Male	197 (64.59)	48 (67.61)
Transplant Diagnosis (n, %)		
Chronic Obstructive Pulmonary Disease (COPD)	129 (42.30)	24 (33.80)
Interstitial Lung Disease (ILD)	115 (37.70)	32 (45.07)
Cystic Fibrosis (CF)	44 (14.43)	10 (14.08)
Bronchiectasis	9 (2.95)	3 (4.23)
Other	8 (2.62)	2 (2.82)
CMV Mismatch Status (n, %)		
R+	255 (83.61)	54 (76.06)
D+/R−	40 (13.11)	15 (21.12)
D−/R−	10 (3.28)	2 (2.82)

**Table 2 medsci-12-00056-t002:** Type of infection according to aetiological agent.

Type of Infection and Microorganism	Frequency (%, N)
**Bacterial**	**56.25 (9)**
*Staphylococcus aureus*	12.50 (2)
*Campylobacter jejuni*	6.25 (1)
*Pseudomonas aeruginosa*	6.25 (1)
*Enterecoccus faecium*	6.25 (1)
*Klebsiella pneumoniae*	6.25 (1)
*Haemophilus influezae*	6.25 (1)
*Clostridium difficile*	6.25 (1)
*Mycobacterium tuberculosis*	6.25 (1)
**Fungal**	**25.00 (4)**
*Aspergillus* spp.	12.50 (2)
*Candida albicans*	6.25 (1)
*Penicillium* sp.	6.25 (1)
**Viral**	**18.75 (3)**
Influenza	12.50 (2)
Cytomegalovirus	6.25 (1)

## Data Availability

Data are contained within the article.

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
