# Peer review of "Impact of Neutropenia on Clinical Outcomes after Lung Transplantation"

_medsci, 2024, doi:10.3390/medsci12040056_

Round 1
Reviewer 1 Report
Comments and Suggestions for Authors
Comments to the Author:
In their manuscript, the authors and colleagues conducted a retrospective analysis of data on lung transplant recipients (LTR) and their outcomes, including neutropenia frequency, GCSF treatment within 18 months post-transplant, and its association with rejection, CLAD, overall survival, and infection development. However, I have outlined my concerns below.
- In lines 96-99 of the results section, the author found 71 patients (23.28%) were treated with GCSF. The authors should provide details about the number of patients from different categories of neutropenia (mild, moderate, severe) included in the 71 patients for treatment with GCSF.
- Conducting a thorough survival curve analysis for all three categories of neutropenia (mild, moderate, severe) that were treated with GCSF vs. No GCSF treatment is crucial. This comprehensive analysis will provide a clearer view of whether GCSF treatment is a direct cause of mortality or if complications associated with neutropenia and transplantation are the primary contributors to mortality.
- The authors should add two more columns to Table 1. The first column should provide the number of patients with neutropenia in each category, and the second column should provide the number of patients treated with GCSF.
- Please correct Table 1 and Table 2: The percentages should be present up to two decimal points separated by a point, not by a comma. Hence, remove the comma and replace it with a point to avoid confusion.
- The authors referenced study 9, which has provided significant information about the association between LTR, neutropenia, and survival. Their findings contradicted the results of the current studies, with the authors attributing the difference to follow-up time. Therefore, it is necessary for the authors to substantiate their results with previous scientific literature and explain why these differing results are attributed to variations in follow-up time in discussion. Please incorporate the follow-up time from previous studies when referencing this literature in the discussion. This will give the audience a clear perspective and enable them to compare the results. Discussion can be elaborated on the previous scientific findings and their relevance to the present study.
Author Response
1. In lines 96-99 of the results section, the author found 71 patients (23.28%) were treated with GCSF. The authors should provide details about the number of patients from different categories of neutropenia (mild, moderate, severe) included in the 71 patients for treatment with GCSF.
Thank you for pointing this out. We agree with this comment. Of the 71 patients who received GCSF, only 18 had mild neutropenia, while 53 had moderate/severe neutropenia. We do not have the breakdown of patients with moderate and severe separately. We have included these data on the page 3, paragraph 4 and line 115, and add this comment “Regardless of its severity GCSF administration was prescribed more frequently in patients prone to bacterial or fungal infections after lung transplantation".
2. Conducting a thorough survival curve analysis for all three categories of neutropenia (mild, moderate, severe) that were treated with GCSF vs. No GCSF treatment is crucial. This comprehensive analysis will provide a clearer view of whether GCSF treatment is a direct cause of mortality or if complications associated with neutropenia and transplantation are the primary contributors to mortality.
Thank you for pointing this out. We found no significant differences in overall survival in terms of severity of neutropenia per patient (p= 0.2418). In order to remark this aspect we add the survival curve of patients in relation to neutropenia severity. Page 4, Figure 2.
3. The authors should add two more columns to Table 1. The first column should provide the number of patients with neutropenia in each category, and the second column should provide the number of patients treated with GCSF.
Thank you for pointing this out. We have collaborated to complete the data requested by the reviewers. We have added one more column with patients treated with GCSF in table 1. Neutropenia has been developed at different times post-transplantation and therefore we have not analysed patient characteristics according to severity.
4. Please correct Table 1 and Table 2: The percentages should be present up to two decimal points separated by a point, not by a comma. Hence, remove the comma and replace it with a point to avoid confusion.
Agree. We have changed commas to points in both tables to improve comprehension.
5. The authors referenced study 9, which has provided significant information about the association between LTR, neutropenia, and survival. Their findings contradicted the results of the current studies, with the authors attributing the difference to follow-up time. Therefore, it is necessary for the authors to substantiate their results with previous scientific literature and explain why these differing results are attributed to variations in follow-up time in discussion. Please incorporate the follow-up time from previous studies when referencing this literature in the discussion. This will give the audience a clear perspective and enable them to compare the results. Discussion can be elaborated on the previous scientific findings and their relevance to the present study.
Thank you for pointing this out. Reference 9 is related to heart transplant recipients. We compared our result in lung transplant patients with those reported by Tague et al. (reference 8). The comparation is very difficult because these authors did not specify any time point of follow-up, while our analysis was limited to the first 18 months post-transplant. In fact, this is one of the limitations of our study already described in the paper because our follow-up period was limited to the first 18 months post-surgery. Although this period is the one with the highest risk of post-transplant complications, the interval analysed may be insufficient for a complete and extrapolable assessment of long-term outcomes.
Reviewer 2 Report
Comments and Suggestions for Authors
This article provides valuable insights into the incidence of neutropenia following lung transplantation, the proportion of cases requiring GCSF administration, and its correlation with prognosis. The content is well-organized, and the data presented offer practical value for clinical management. However, I suggest that the conclusions could be made more robust with further investigation into several factors, such as the specific reasons necessitating GCSF treatment, the timing of GCSF administration, adjustments to immunosuppressive therapy during GCSF treatment, and the relationship between the frequency of GCSF administration and patient prognosis. Therefore, it is considered desirable to add explanations for the following comments .
Comments:
1) The criteria for GCSF administration should be clearly defined.
2) It would be useful to briefly explain when GSF administration is most likely to be required after transplantation.
3) It is important to explain the institution's strategy for adjusting immunosuppressive therapy in response to neutropenia.
4) To better understand the higher mortality rate among cases receiving GCSF, it is necessary to clearly state the causes of death and describe the relationship between the number of GCSF administrations and mortality.
Author Response
1. The criteria for GCSF administration should be clearly defined.
Thank you for pointing this out. We agree with this comment. Therefore we have included the criteria for prescribing neutropenia that are based on the severity of the neutropenia, the patient's characteristics and the physician's judgment (page 2, paragraph 10 and line 83).
2. It would be useful to briefly explain when GCSF administration is most likely to be required after transplantation.
Agree. Accordingly, we have added these parts:
"We found no significant differences in the occurrence of neutropenia taking into account the time post-transplantation in the different periods." (page 3, paragraph 2 and line 104). (Attached figure)
"Regardless of its severity, GCSF administration was prescribed more frequently in patients prone to bacterial or fungal infections after lung transplantation." (page 3, paragraph 4 and line 116).
3. It is important to explain the institution's strategy for adjusting immunosuppressive therapy in response to neutropenia.
Thank you for pointing this out. We agree with this comment. Therefore we have explained it in Material and Methods:
"Immunosuppression management was adjusted based on the patient's characteristics and the balance between the risk of rejection or infection. Mycophenolate was reduced or temporarily discontinued, while the usual doses of tacrolimus and corticosteroids were maintained. In relation to antimicrobial prophylaxis, sulfamethoxazole/trimethoprim and valganciclovir were also discontinued, and we used as an altermnative for CMV prophylaxis the specific CMV hyperinmmunoglobulin". (Page 2, paragraph 9 and line 77).
4. To better understand the higher mortality rate among cases receiving GCSF, it is necessary to clearly state the causes of death and describe the relationship between the number of GCSF administrations and mortality.
Thank you for pointing this out. We analysed the causes of mortality in the 18 patients who received GCSF and died at the end of follow-up. Infections were the causes of dead in 10 patients (55.6%), progressive CLAD in 5 (27.78%), humoral rejection in 2 (11.11%) and massive hemoptisis in 1 (5.56%). (Page 3, paragraph 6 and line 126).
The higher mortality of our patients treated with GCSF could be related to ajustments in immunosuppression and infection prophylaxis. In this regard, the relationship between the number of units received and mortality had not been considered in the inital statistical analysis.
Round 2
Reviewer 1 Report
Comments and Suggestions for Authors
Most of the concerns raised by the reviewers have been addressed, and The manuscript has been significantly improved.